# Evaluation of Phase Transformation and Mechanical Properties of Metastable Yttria-Stabilized Zirconia by Nanoindentation

**DOI:** 10.3390/ma12101677

**Published:** 2019-05-23

**Authors:** Ningning Song, Ziyuan Wang, Yan Xing, Mengfei Zhang, Peng Wu, Feng Qian, Jing Feng, Longhao Qi, Chunlei Wan, Wei Pan

**Affiliations:** 1State Key Lab of New Ceramics and Fine Processing, School of Materials Science and Engineering, Tsinghua University, Beijing 100084, China; songnn16@mails.tsinghua.edu.cn (N.S.); wang-zy15@mails.tsinghua.edu.cn (Z.W.); Xingy14@mails.tsinghua.edu.cn (Y.X.); zhangmf15@mails.tsinghua.edu.cn (M.Z.); lhqi@tsinghua.edu.cn (L.Q.); wancl@tsinghua.edu.cn (C.W.); 2Faculty of Material Science and Engineering, Kunming University of Science and Technology, Kunming 650093, China; peng.w.st@foxmail.com (P.W.); qf18487267689@163.com (F.Q.); jingfeng@kmust.edu.cn (J.F.)

**Keywords:** Yttria-stabilized zirconia, phase transformation, nanoindentation, thermal barrier coatings

## Abstract

Microscopical nonuniformity of mechanical properties caused by phase transformation is one of the main reasons for the failure of the materials in engineering applications. Accurate measurement of the mechanical properties of each phase is of virtual importance, in which the traditional approach like Vickers hardness cannot accomplish, due to the large testing range. In this study, nanoindentation is firstly used to analyze the mechanical properties of each phase and demonstrate the phase transformation in thermal barrier coatings during high-temperature aging. The distribution of T-prime metastable tetragonal phase, cubic and tetragonal phase is determined by mapping mode of nanoindentation and confirmed with X-ray diffraction and scanning electron microscope observation. The results show that during 1300 °C aging, the phase transition of metastable Yttria-Stabilized Zirconia induces the quick decrease of T′ phase content and an increase of T and C phases accordingly. It is found that there are some fluctuations in the mechanical properties of individual phase during annealing. The hardness and Young’s modulus of T′ increase at first 9 h, due to the precipitation of Y^3+^ lean T phase and then decrease to a constant value accompanied by the precipitation of Y^3+^ rich C phase. The relevant property of C phases also increases a little firstly and then decreases to a constant, due to the homogenization of Y^3+^ content, while the hardness and Young’s modulus of T phase remain unchanged. After aging of 24h the hardness of T′, C and T phases are 20.5 GPa, 21.3 GPa and 19.1 GPa, respectively. The Young’s modulus of T′, C and T phases are 274 GPa, 275 GPa and 265 GPa, respectively. Present work reveals the availability of nanoindentation method to demonstrate the phase transformation and measure mechanical properties of composites. It also provides an efficient application for single phase identification of ceramics.

## 1. Introduction

In order to prolong gas turbine lifetime and increase operation temperature, thermal barrier coatings (TBCs) are required to protect the hot parts of the surface from high temperature, high pressure and corrosive gases [1,2]. 8 wt.% Yttria-stabilized zirconia (8YSZ) has been effectively applied as TBCs [3,4], exhibiting low thermal conductivity [2,5,6], superior mechanical properties and high thermal expansion coefficient [7]. The coatings are primarily fabricated by air plasma spray (APS) or electron beam physical vapor deposition (EB-PVD) methods [8], in which the T-prime metastable tetragonal (T′) phases of YSZ are mainly formed. According to the phase diagram of ZrO_2_-Y_2_O_3_ [9], the T′ phase containing 8 wt.% Yttria does not exist as an equilibrium phase in a solid state. However, it usually occurs in the non-equilibrium fabrication processes, like APS or EB-PVD [8]. When the operating temperature is higher than 1250 °C, the metastable T′ phase in TBCs would decompose into thermodynamically stable tetragonal (T) phase and cubic (C) phase accompanied by Y ion diffusion [10,11,12]. T phase has a less concentration of Y element than T′ phase while the C phase has a higher one than T′ phase. The newly generated T phase further transforms into monoclinic (m) phase during thermal cycling, accompanied by 5–7% volume expansion [13]. Previous work [11,12,13] reported the kinetics of the phase transforming from T′ to T and C phase, and the Y^3+^ ion and oxygen vacancy behaviors during the phase transformation were identified by Raman signal [14]. The transformation from T′ phase to T and C phases would result in the degradation of mechanical properties [11]. Since the 1990s, there have been many researchers proving that T′ phase has a ferroelastic toughening mechanism, and the existence of T′ could toughen the ceramic by re-orientating the c axis in the crystal cells and absorbing elastic energy to slow down the cracking [15,16,17,18]. Normally, T′ phase shows better mechanical properties than C phase, due to its reversible ferroelastic domain switching [14,19,20].

Therefore, it is essential to monitor the changes in the mechanical properties in each phase during phase transformation [21,22,23]. However, it is difficult to obtain ultrapure phase samples because phase transformation is inevitable [15]. Moreover, as the grains of T′, T and C phases are very fine (the maximum grain size of T′ and C phases is 4–5 μm, while that of T phase is about 300–600 nm), the indenter of regular Vickers hardness measurement is much larger (over ten microns) than the grain. Therefore, the hardness and Young’s modulus measured using Vickers indenter could not reflect the properties of single-phase grain [16]. It is also hard to distinguish the properties of a single phase in the matrix [17]. Hence, the interaction of phases, the influence of phase composition and defects on mechanical properties cannot be discussed by means of regular Vickers hardness measurement.

Recently, the mechanical properties of single phase or individual grains in composite materials, which is small size even nanoscale, could be evaluated by nanoindentation technique [18,24,25,26]. Hay et al. investigated the influence of the substrate on the evaluation of mechanical properties in thin films using instrumented nanoindentation [27]. Recent work regarding the constructing two-dimensional nanoindentation mapping, combined with theoretical calculations, has been reported for the evaluation of mechanical properties’ distribution in Fe-6.0W-5.0Mo-4.0Cr-2.0V-0.83C alloy by Chong et al. [28]. Hardness and Young’s modulus of different precipitated phases in the alloy were measured using nanoindenter. Amanieu et al. distinguished compact particles from the epoxy matrix using a selective statistical nanoindentation method and measured the hardness and Young’s modulus of LiMn_2_O_4_ in the composite cathode [29]. On the other hand, Mughal et al. studied the fracture toughness of Li_x_Mn_2_O_4_ cathode under different states of charge by a high-speed nanoindentation mapping [30], measuring and processing the mechanical properties of large data points in a small period of acquisition time. Although atomic force microscopy (AFM) provides better spatial resolution and extracts the elastic modulus by contact mode [31], it is more difficult to measure mechanical properties for a stiff sample and property distribution in micron scale range, thereby showings higher uncertainty than nanoindentation [32,33].

In the research on thermal barrier coatings, nanoindentation has been widely used in yttria stabilized zirconia materials. Bolelli et al. reported a work using high-throughput nanoindentation to detect the nano-mechanical properties in TBCs during thermal cycling [34]. Some researchers have detected the mechanical properties of coarse equiaxed grains and fine columnar one in as-deposited T′ phase YSZ top coatings using nanoindentation system [34]. Nath et al. applied this method to test the variation of hardness and Young’s modulus in YSZ based TBC before and after isothermal aging [35]. Mao et al. analyzed the reverse indentation size effect on 8 mol% YSZ by Berkovich nanoindentation under ultra-low load [36]. However, until now, there has been no report investigating the mechanical properties of individual T′, T and C phases in YSZ used for TBCs through nanoindentation.

In this research, we aim to study the mechanical properties and phase distribution in 8YSZ composite during phase transformation along with high-temperature annealing using nanoindentation techniques. High T′ phase content 8YSZ powders were prepared by APS, and densified by spark plasma sintering (SPS). Specimens with different contents of T′, C and T phases were prepared by annealing at 1300 °C. The hardness and Young’s modulus of each phase were evaluated by continuous stiffness measurement mode, and surface phase distribution exhibited through 3D mapping mode of nanoindenters. The variations in mechanical properties for each phase along with high-temperature aging were investigated and explained.

## 2. Materials and Methods

### 2.1. Specimen Preparation

In order to obtain the T′ phase 8 wt.% 8YSZ, commercial 8YSZ ceramic particles were directly sprayed into the cooling water with APS unit (GTV-MF-P-HVOF-K-ARC, F6 spraying gun, GTV, Betzdorf, Germany) [37]. Then the obtained powders were ball-milled for 8 h and sintered at 1450 °C for 5 min under the condition of unidirectional pressure 50 MPa by the spark plasma sintering method (Dr. Sinter 1020 SPS, Sumitomo Coal Mining Co., Tokyo, Japan) [11] to get highly densified T′ phase specimens. Then the sintered specimens were annealed in Nabertherm furnace at 1300 °C for 2 h, 9 h, 12 h and 24 h, respectively. The sample surface was machined, polished and cleaned by etching using Argon ion beam for 30 min, 5 Kev (Precision etching and coating systems PECSII 685, Gatan, UK).

### 2.2. Structure and Morphology of 8YSZ

The morphology of annealed specimens after aging for 2 h, 9 h, 12 h and 24 h at 1300 °C were observed with field-emission scanning electron microscopy (FE-SEM, Merlin Compact, Zeiss, Jena, Germany) with the accelerating voltage of 8 kV. Besides, Energy Dispersive Spectroscopy (EDS, Oxford Instrument, Abingdon, Oxfordshire, UK) was investigated to distinguish the element content among grains in mapping and line scanning mode with the accelerating voltage of 15 kV. EDS data was processed using Aztec Energy software to obtain quantitative analysis results. The phase composition was characterized by XRD (D/max-2500, Rigaku, Tokyo, Japan) at a scanning speed of 0.2°·min^−1^ to differentiate T (004), T′ (004), C (400), T′ (400) and T (400) diffraction peaks in the 2θ range of 72.5°–75.5° [37].

### 2.3. Mechanical Properties of 8YSZ

The mechanical properties of 8YSZ were measured with a nanoindenter (Nanomechanics, iMicro, Oak Ridge, TN, USA), and nanoindentation mapping was conducted through the NanoBlitz three-dimensional mechanical property mapping mode. The mapping area was 150 µm × 150 µm with 900 points and the load was 5 mN. The Poisson’s ratio of 8YSZ was assumed to be 0.25 [34]. Besides, the mechanical properties of single-phase grains were also measured through ten valid tests with a Berkovich diamond indenter by Continuous Stiffness Measurement (CSM) technique. Load segment was conducted under the conditions of a constant strain rate of 0.05 s^−1^, a dynamic frequency at 75 Hz, and a maximum penetration depth of 100 nm. In order to ensure the accuracy of data, nanoindentation instrument was calibrated before and after testing with a certified fused silica sample. Nanoindentation mapping is a high-speed nanoindentation method, with each indentation corresponding to hardness and Young’s modulus value from the maximum depth. Meanwhile, continuous stiffness measurement mode promises to display the hardness and Young’s modulus as a function of depth, due to its harmonic force.

## 3. Results and Discussion

### 3.1. Morphology and Y Element Distribution of 8YSZ

APS 8YSZ mainly contains metastable T′ phase, but it decomposes into Y rich C phase and Y lean T phase at a high temperature [38,39]. The surface morphology and corresponding content of Y distribution were observed through FE-SEM and EDS, as shown in Figure 1a,b, respectively. The nanoscale grains contain less than 4 wt.% Y element, the content of Y in coarse grains is higher than 8 wt.%, and the content of Y in other grains is about 5–6 wt.%. The fine grains with lean Y content are T phase and the coarse grains with rich Y content are C phase, which is consistent with the result in the literature [11]. Line-scanning of elements Y, Zr and O under the SEM clearly demonstrates the inhomogeneous distribution of Y in the grains, as shown in Figure 1c,d. The line scanning results also reveal the coarse C phase grains and fine T phase grains in the samples.

Figure 2 exhibits the surface morphology of the samples annealed at 1300 °C for 2 h, 9 h, 12 h and 24 h, respectively. On the base of previous research, the small grains with nanoscale diameter represent the T phase, the Y rich phase with 3–4 μm diameters is C phase and the others are T*ʹ* phase, as marked in Figure 2. In the initial state of 2 h thermal treatment, T*ʹ* phase is the main component in the 8YSZ bulk with the appearance of a small amount of T phase, implying that the phase transformation process just begins. As the aging time is prolonged, C phase gradually nucleates, grows and substitutes more T*ʹ* phase. It is clear that the T and C phases increase while T*ʹ* phase reduces significantly with the increase of annealing time.

### 3.2. Phase Distribution on 8YSZ Surface by Nanoindentation Mapping

During high-temperature aging, phase transformation results in changes in the mechanical properties in 8YSZ. Newly formed C phase has enhanced the hardness and modulus, due to the alloy-effect of high content Y doping, while T phase exhibits soft characteristics on account of lower concentration of Y and its larger tetragonality than metastable T′ phase [40]. The hardness and modulus distribution of different phases in 8YSZ are recorded, as shown in Figure 3 and Figure 4. Since the size of phase grains is smaller than 5 μm and even nanoscale, it is hard to distinguish three phases just by visual inspection. The color distribution resulted from minor changes in mechanical properties among the three phases still exhibits some information. Figure 3 shows the hardness distribution on the surface of 8YSZ samples annealed at 1300 °C for 2 h, 9 h, 12 h and 24 h, respectively. In Figure 3a the imaging is covered in orange and yellow color, indicating the hardness value about 20–30 GPa. Much more blue areas about 17 GPa are distributed in Figure 3b. It is clear that as the annealing time prolongs, the soft phase area (marked as dark blue) increases at the initial stage, and then becomes almost unchanged, indicating the formation of the T phase. Meanwhile, the hard phase area (marked as dark red and green) sees enhancement accompanied by the appearance of the C phase as Figure 3c. As shown in Figure 4, the sky blue and dark blue area are clearly generated after heat treatment. Young’s modulus also varies with the annealing time, in which both of the low and high Young’s modulus phase areas increase, leading to larger variation within the whole area. Thus, Figure 3 and Figure 4 clearly demonstrate the distribution of each phase by nanoindentation mapping, providing a better description of the whole transformation process.

### 3.3. Phase Transformation Analysis of 8YSZ

This phase degradation of the APS 8YSZ is also detected using high-resolution XRD scanned from 72.5° to 75.5° (2θ) [41]. The volume fraction of each phase is determined using the relevant phase intensity I_i_ (i = T’, T, C, M) by following Equations [42,43].
(1)$$V_{T}=\frac{\left(1-V_{M}\right)\times \left(I_{T}\left(400\right)+I_{T}\left(004\right)\right)}{I_{T}\left(400\right)+I_{T}\left(004\right)+I_{C}\left(400\right)+I_{T^{\prime }}\left(400\right)+I_{T^{\prime }}\left(004\right)}$$
(2)$$V_{T^{\prime }}=\frac{\left(1-V_{M}\right)\times \left(I_{T^{\prime }}\left(400\right)+I_{T^{\prime }}\left(004\right)\right)}{I_{T}\left(400\right)+I_{T}\left(004\right)+I_{C}\left(400\right)+I_{T^{\prime }}\left(400\right)+I_{T^{\prime }}\left(004\right)}$$
(3)$$V_{M}=\frac{I_{M}\left(11\bar{1}\right)+I_{M}\left(111\right)}{I_{M}\left(11\bar{1}\right)+I_{M}\left(111\right)+I_{C,T}\left(111\right)}$$
(4)$$V_{C}=1-\left(V_{M}+V_{T}+V_{T^{\prime }}\right)$$
where, V_M_, V_T_, V_T′_ and V_C_ are the volume fraction of the four phases and I_i_ is the integrated intensity of each phase peaks. Here the diffraction peak (111) corresponding to M phase is too weak to be observed [44], and the other five diffraction peaks C (004), T (004), T (400), T′ (004) and T′ (400) are differentiated by Lorentz peak fitting (R^2^ > 0.99), as shown in Figure 5. Without heating treatment sample contains T’ phase only. And the resulting phase volume content changes are shown in Figure 6b. In XRD results, as mentioned above, the T′ phase content decreases, due to the precipitation of C and T phases, where T phase can easily precipitate from the T′ phase at the initial annealing stage, due to a similar structure of T and T′ phases. In contrast, the C phase content is relatively low at the initial stage, due to the high Y requirement for nucleation. Compared with the T phase, it is more difficult to form the C phase. Then, the C phase content shows a moderate increase in the initial 9 h annealing, and rises rapidly in the subsequent heat treatment process. The area of T′, T and C phase versus annealing time are statistically analyzed using the data of nanoindentation mapping, as shown in Figure 6a. The data are processed by Matlab software. The criteria are identified by the hardness value or corresponding color coding at different annealing time. Meanwhile, the change in mechanical properties for the individual phase is also to be considered. The nanoindentation mapping (Figure 6a) and XRD (Figure 6b) results show a similar trend, representing the accuracy of nanoindentaion mapping results.

### 3.4. Mechanical Properties of Single Phase

To obtain hardness and Young’s modulus of individual grains in T′, T and C phases, nanoindenter with CSM technique is adopted, as shown in Figure 7. Although C phase has already precipitated during first 2 h heat treatment, it could not be detected by the indentation clearly in present tests, due to the small size of C grains, whereas it could be found after 9 h aging owing to the grain growth in spite of the increase of amount of C phase is not so significant compared with that after 2 h aging. As a very small amount of C phase in the sample can be found during the initial 9 h, its hardness and Young’s modulus are measured only for the samples annealed for a longer time.

Figure 8a,b exhibits the hardness and modulus of the three phases, respectively. 10 nanoindentation tests for each phase are performed and standard deviation bars are marked in Figure 8. The results suggest that, there are standard deviation overlaps for 9 h, 12 h and 24 h, particularly for T′ and C phases. This may due to the uncertainty of the indentation position during testing, and some values come from the grain boundary of T′ and C phases. The values of hardness and elastic modulus of T′ phase are about 20.5–22.5 GPa and 270–295 GPa, respectively, as measured in this research with a nanoindentor. These data are a little bit higher than that reported in the literature (approximate 20 GPa and 245 GPa) [34], which may be attributed to the difference in loading mode (constant depth of 100 nm with the load of 5 mN in this research, while 3 mN in the literature).

Furthermore, the mechanical properties vary with annealing time, and the changing trend of hardness is similar to the Young’s modulus for the same phase. The mechanical properties of T′ and C phases both first see enhancement and reach a maximum at 9 h and 12 h respectively, and then slide down to a constant value, while the hardness and modulus of T phase remain unchanged. At the first stage, enhancement of the mechanical properties of T′ phase may arise from the richness of Y content, caused by the precipitation of T grains (lean Y phase). When the Y concentration in T′ grains reaches to a critical point, C grains begins to precipitate, and through the diffusion from T′ grains, Y content in C grains may increase, leading to enhanced hardness and modulus of C phase. As the Y concentration in C grains reaches equilibrium, the hardness and modulus become unchanged. In the case of T phase grains, the mechanical properties are almost a constant during the whole annealing, which is possibly due to the very fine grain size and rapid equilibrium of the Y concentration. Generally, the C phase has relatively higher hardness and modulus than T′ and T phases owing to the nature of a cubic structure.

The results show that nanoindentation has successfully extracted the mechanical properties of each phase in the 8YSZ composites, and the properties have a close relationship with the crystal structure and element content varies with phase transition. Meanwhile, the results of the three phases are also comprehensively reflected in the nanoindentation mapping.

## 4. Conclusions

In summary, the high content of metastable tetragonal T′ phase YSZ samples were prepared by spark plasma sintering of air plasma sprayed 8YSZ powders. The YSZ samples containing T′ phase were aged at 1300 °C, and the precipitation of cubic phase (C) and tetragonal phase (T) and their content changing along with aging time were confirmed using XRD and EDS. The hardness and Young’s modulus of individual phase grain were measured by a nonoindentation. The results show that the hardness and Young’s modulus of C phase are higher than T′ phase and the T phase shows the least value after 24 h heating treatment. C phase has the highest Y content and T phase has the lowest Y content, corresponding to their mechanical properties. Nonoindentation mapping on the surface of samples clearly shows the distribution of these phases and their changing with aging time. The values of the hardness and Young’s modulus of the samples containing different phases showed obvious inhomogeneity, as demonstrated in nonoindentation mapping. Moreover, the separated T′, T and C phase area can be easily counted by the nonoindentation, which is an easy method to test phase transition and evaluate phase content changing. Ultimately, their validity is confirmed by means of XRD, EDS and SEM analysis. This research creatively introduces nanoindentation as a new approach to study phase transition and mechanical properties of a single phase in ceramic materials.

## Figures and Tables

**Figure 1 materials-12-01677-f001:**
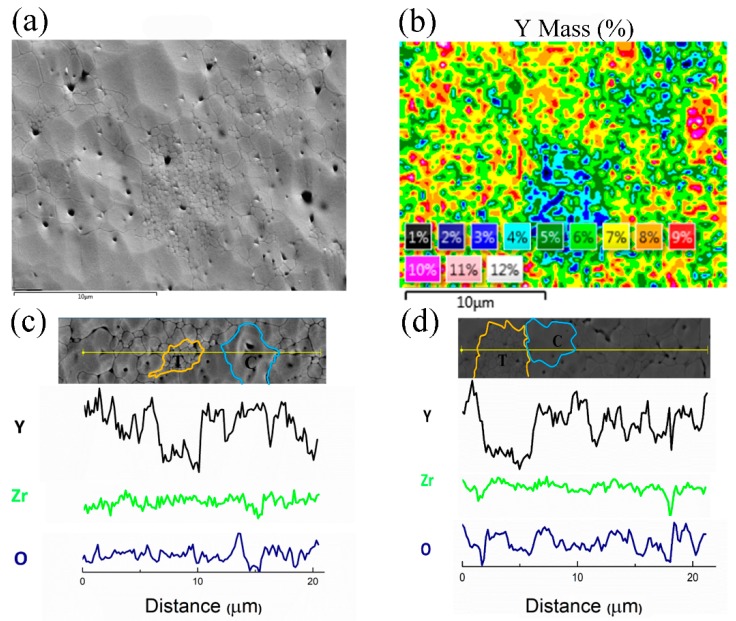
EDS mapping and a line graph of Y element on the surface of 8YSZ annealed at 1300 °C for 24 h. (**a**) SEM image (**b**) corresponding EDS Mapping, different color coding indicating the various weights of the Y element (**c**) and (**d**) EDS line scanning of another two different areas.

**Figure 2 materials-12-01677-f002:**
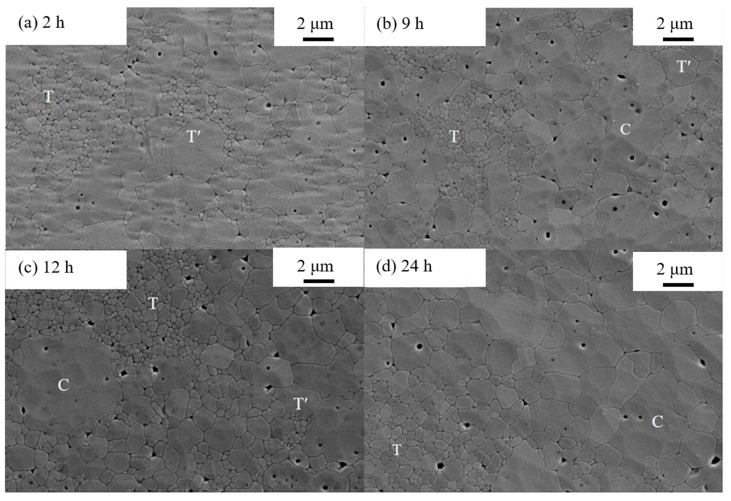
SEM images of 8YSZ morphology with periods of annealing time (**a**) 2 h, (**b**) 9 h, (**c**) 12 h, (**d**) 24 h.

**Figure 3 materials-12-01677-f003:**
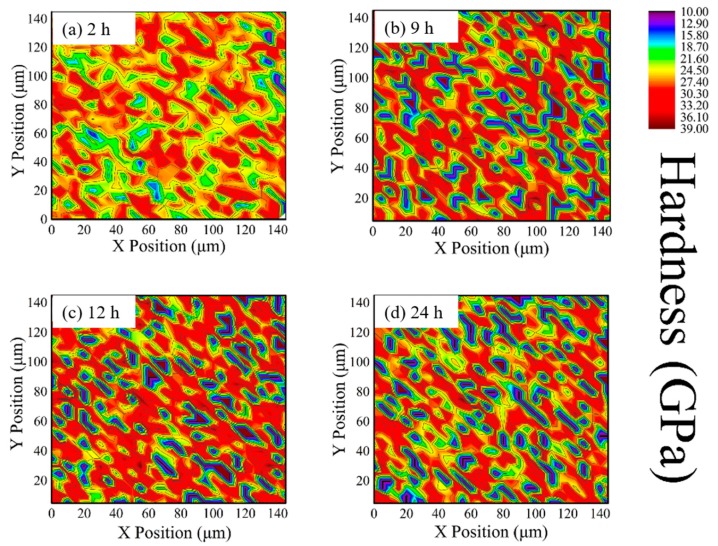
Plane projections of the hardness of 8YSZ with the annealing time (**a**) 2 h, (**b**) 9 h, (**c**) 12 h, (**d**) 24 h. The right color scale indicates hardness variation.

**Figure 4 materials-12-01677-f004:**
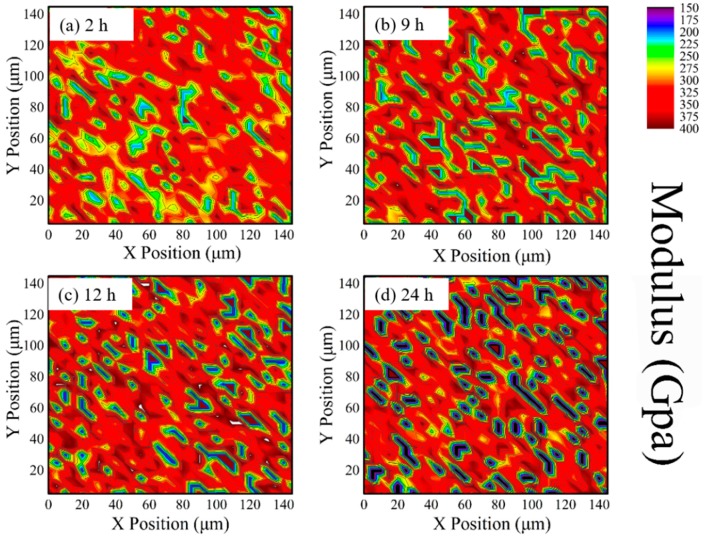
Plane projections of Young’s modulus of 8YSZ with annealing time (**a**) 2 h, (**b**) 9 h, (**c**) 12 h, (**d**) 24 h. The right color scale indicates Young’s modulus variation.

**Figure 5 materials-12-01677-f005:**
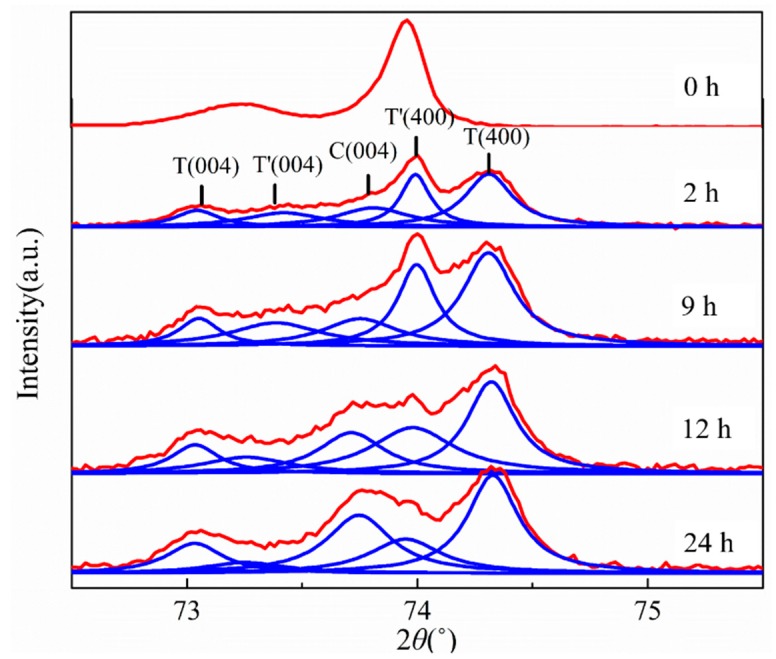
The evolution of XRD patterns and five diffraction peaks of C (004), T (004), T (400), T′ (004) and T′ (400) by Lorentz peak fitting (R^2^ > 0.99).

**Figure 6 materials-12-01677-f006:**
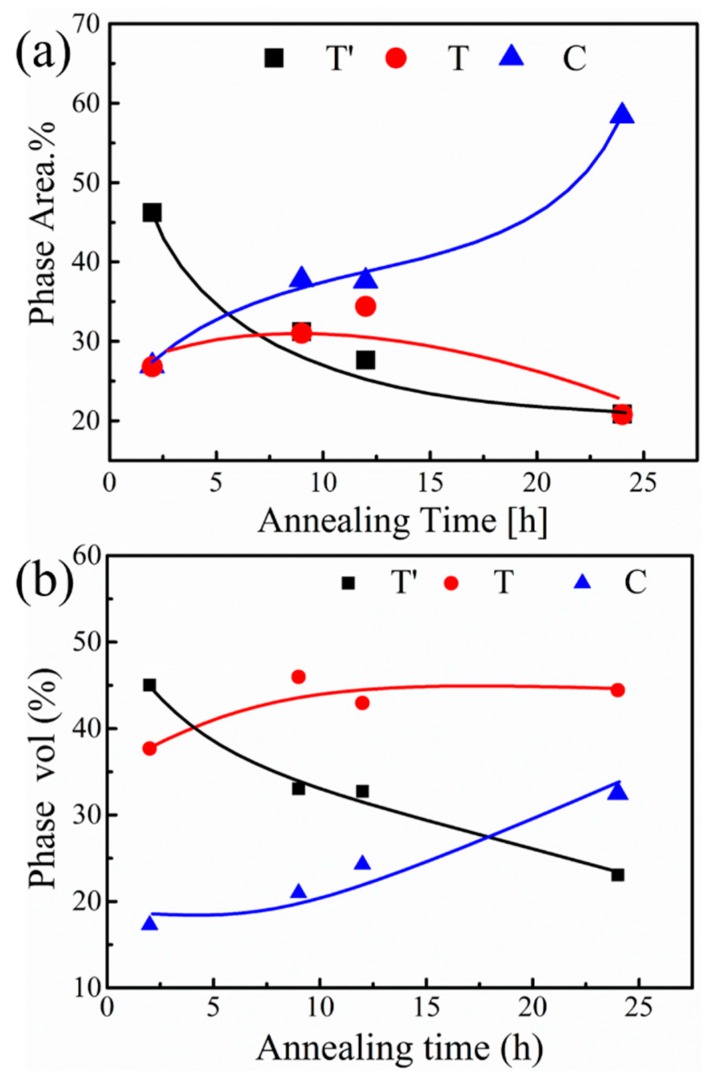
(**a**) The areas fraction of T′, T and C phases on the surface from the data of nanoindentation mapping. (**b**) Phase content of 8YSZ as a function of annealing time from XRD results.

**Figure 7 materials-12-01677-f007:**
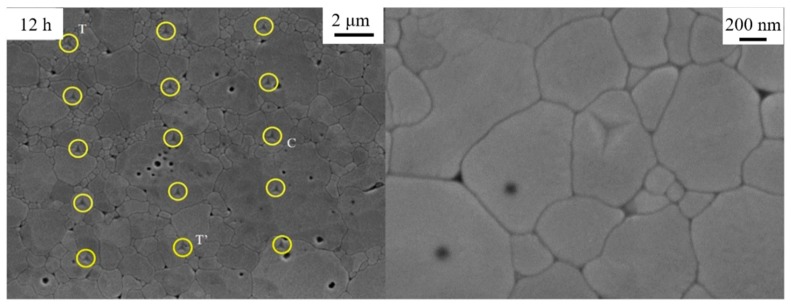
Nanoindentation of 8YSZ microstructure with annealing time 12 h.

**Figure 8 materials-12-01677-f008:**
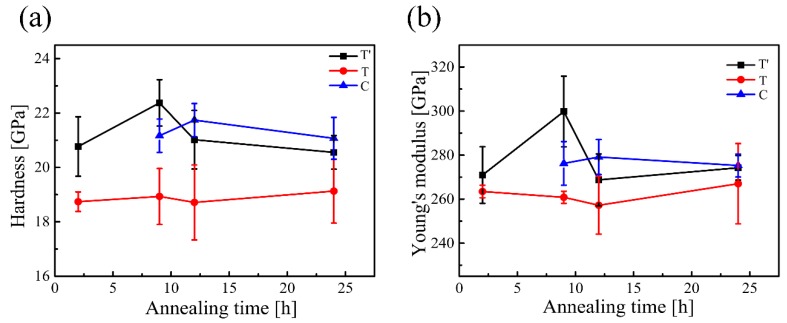
(**a**)Nano-indentation hardness (**b**)Young’s modulus as a function of annealing time.

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
