# Peer review of "Evaluation of Phase Transformation and Mechanical Properties of Metastable Yttria-Stabilized Zirconia by Nanoindentation"

_materials, 2019, doi:10.3390/ma12101677_

Reviewer 1 Report

The paper reports the use of instrumented nanoindentation to evaluate the mechanical properties of different phases in yttria-stabilized zirconia. My comments are as follows:  

Introduction

The introduction is very short and there are significant aspects which are not discussed, few examples to improve it are as follows:

1/ Authors mentioned T, T’ and C phases during the coating formation, however there is very less detail on how these phases are formed. This transformation is a well-known fact in the literature, however it should be discussed in more detail.

2/ Authors mentioned that the Vickers hardness testing is not a suitable method as it provides the average value. What is meant by AVERAGE vale? They should highlight in detail with examples from literature and clearly differentiate the reasons (macro scale testing vs micro/nano scale testing). Also, the size (mention what scale range you are talking about) of the individual phases is different and hence nanoindentation is more suitable to explore the individual mechanical properties as compared to the Vickers hardness testing.

 3/ The section corresponds to nanoindentation mapping should be discussed in detail. There are so many examples in the literature about the versatility of this technique (small acquisition time, large data (indents) points, ease of data processing etc.). It should also be discussed that the application of this technique to wide range of materials. Few recent articles are as follows:

https://doi.org/10.1016/j.matdes.2019.107615

https://doi.org/10.1016/j.scriptamat.2016.01.023

https://doi.org/10.1016/j.msea.2013.11.044

4/ The authors should clearly state how the approach taken in this article is different than literature. Are there any reports which used the nanoindentation method to evaluate the mechanical properties on individual phases particularly for yttria stabilized zirconia? If yes, then how this work differs from them? Few examples in which nanoindentation is used to investigate the mechanical properties is as follows. Authors should clearly mention how their work is different from the already published work.

https://doi.org/10.1016/j.matdes.2019.107615 

https://doi.org/10.1016/j.ceramint.2014.11.039

https://doi.org/10.1016/j.apsusc.2015.02.108

Materials and Method

1/ Please provide imaging condition for SEM as well as for EDS spectra.

2/ Please add how many tests were performed for single phase grain tests and how the instrument was calibrated? It is a usual practice to run fused silica sample before and after the test and it should be mentioned here. More details on how to structure this part can be found in the following reference.

 https://doi.org/10.1016/j.matdes.2019.107615

Results and discussion

This section needs significant improvement. I think the overall structure of the article is very confusing. The section starts with the E and H maps which do not give any idea where those indentations were performed and what is meant by those different colours. A reader will find it very confusing to understand!

Here are some suggestions that might improve the overall presentation.

1/ Figure 4 should be the starting point, where one can understand how the phases look like and how the transformation is happening with time.

2/ Phase maps should be the second and complement SEM images. I would prefer that authors highlight the area where the maps were carried out (for details please see reference below). The numbers corresponding to different phases should be highlighted in detail and the discussion should also be made that how the hardness and modulus of different phases changes with the annealing time? I think this is backbone of the article and should be discussed in detail with reference to the literature. The authors should compare the values obtained in this article with the literature values.

https://doi.org/10.1016/j.ijrmhm.2018.04.019

https://doi.org/10.1016/j.matdes.2016.02.087

3/ I do not understand figure 3. The 3c and 3d is perhaps the magnified view from 3a but the chosen area is not highlighted. 3b have different %ages on the figure. What different colour coding means, it is not highlighted in the caption. This should be better presented!

4/ Mechanical properties of single phases (figure 6 and 7, perhaps combine both images?) is very interesting, however discussion should be made about the standard deviation. It is worth highlighting here again that how many indentations were performed and why the standard deviation overlaps for 9, 12 and 24 hours. Detail discussion should be made about the trend.

Conclusions

Based on the above-mentioned changes, this section needs to be re-written with more clarity that how the nanoindentation mapping help distinguish between different phases and provide reliable measurement of both hardness and modulus of different phases.

General Remarks

Technical English needs improvement. There are some grammatical mistakes which authors should carefully check.

Author Response

Dear reviewer:

On behalf of my co-authors, we thank you very much for giving us an opportunity to revise our manuscript. We appreciate for your comments and suggestions on the manuscript entitled “Evaluation of Phase Transformation and Mechanical Properties of Metastable yttria-stabilized zirconia by Nanoindentation (ID: materials-498383)”. We have read the comments carefully and tried our best to revise the manuscript accordingly. Revised portion are highlighted with red words in the paper. The main corrections in the paper and the responses to referee’s comments are as following.

Response to the reviewer 

Introduction

Point#1 Authors mentioned T, T’ and C phases during the coating formation, however there is very less detail on how these phases are formed. This transformation is a well-known fact in the literature, however it should be discussed in more detail.

Response#1 Thank you for your comment. Detail information on phase transformation are added in the introduction.

Point #2 Authors mentioned that the Vickers hardness testing is not a suitable method as it provides the average value. What is meant by AVERAGE vale? They should highlight in detail with examples from literature and clearly differentiate the reasons (macro scale testing vs micro/nano scale testing). Also, the size (mention what scale range you are talking about) of the individual phases is different and hence nanoindentation is more suitable to explore the individual mechanical properties as compared to the Vickers hardness testing.

Response#2 Sorry for being unclear about average value. Because the head of the indenter of regular Vickers hardness measurement is much larger (10~1000 microns) than the grain size, the values measured by Vickers hardness may include the effect of different phases. As the grain size of T′, T and C phases are very fine (the maximums grain size of T′ and C phases are 4~5 mm, and the grain size of T phase is about 300~600 nm), the hardness and Young’s modulus measured using Vickers indenter could not reflect the single phase grain properties. Besides, the radius of nanoindenter is less than 20nm, providing the benefits of exploring the hardness of micro-sized and nonoscale grains. That is why we choose nanoindentation rather than the Vickers hardness. The accurate description is detailed in the article as red words highlighted.

 Point #3 The section corresponds to nanoindentation mapping should be discussed in detail. There are so many examples in the literature about the versatility of this technique (small acquisition time, large data (indents) points, ease of data processing etc.). It should also be discussed that the application of this technique to wide range of materials. Few recent articles are as follows:

https://doi.org/10.1016/j.matdes.2019.107615

https://doi.org/10.1016/j.scriptamat.2016.01.023

https://doi.org/10.1016/j.msea.2013.11.044

Response#3 Nanoindentation mapping really has many applications to explore the mechanical properties of various materials. The revised article already enriched its application and versatility in introduction on account of reviewer’s reference opinion.

Point #4 The authors should clearly state how the approach taken in this article is different than literature. Are there any reports which used the nanoindentation method to evaluate the mechanical properties on individual phases particularly for yttria stabilized zirconia? If yes, then how this work differs from them? Few examples in which nanoindentation is used to investigate the mechanical properties is as follows. Authors should clearly mention how their work is different from the already published work.

https://doi.org/10.1016/j.matdes.2019.107615 

https://doi.org/10.1016/j.ceramint.2014.11.039

https://doi.org/10.1016/j.apsusc.2015.02.108

Response#4 According to the literature given by reviewer, nanoindentation is widely used in the yttria stabilized zirconia. The modification is emphasized in the article with red words. Although some researches deal with the detection of mechanical properties of T′ phase YSZ top coat, and the variation mechanical properties of YSZ coatings after different thermal cycles by nanoindentation, there is no reports investigating the mechanical properties on individual T′, T and C phases for YSZ and their variation along with high temperature aging by nanoindentation.

Materials and Method

Point #1 Please provide imaging condition for SEM as well as for EDS spectra.

Response#1 Thanks for your suggestions. Imaging condition for SEM as well as for EDS spectra was added in the section 2.2.

Point #2 Please add how many tests were performed for single phase grain tests and how the instrument was calibrated? It is a usual practice to run fused silica sample before and after the test and it should be mentioned here. More details on how to structure this part can be found in the following reference.

 https://doi.org/10.1016/j.matdes.2019.107615

Response#2 Thanks for your advice. The mechanical properties of single phase grains were measured containing ten valid tests. In order to ensure the accuracy of data, nanoindentation instrument was calibrated before and after testing on a certified fused silica sample. All details are added in the section 2.3.

Results and discussion

Point #1 Figure 4 should be the starting point, where one can understand how the phases look like and how the transformation is happening with time.

Response#1 Considering the reviewer’s opinion, adjustment of structure in article is completed. Please see the details in the article.

Original Figure 3 and Figure 4 change to Figure 1 and Figure 2, and original Figure 1 and Figure 2 change to Figure 3 and Figure 4. The section 3.1 headings change to Morphology and Y element distribution of 8YSZ.

Point #2 Phase maps should be the second and complement SEM images. I would prefer that authors highlight the area where the maps were carried out (for details please see reference below). The numbers corresponding to different phases should be highlighted in detail and the discussion should also be made that how the hardness and modulus of different phases changes with the annealing time? I think this is backbone of the article and should be discussed in detail with reference to the literature. The authors should compare the values obtained in this article with the literature values.

https://doi.org/10.1016/j.ijrmhm.2018.04.019

https://doi.org/10.1016/j.matdes.2016.02.087

Response#2 Adjustment of images location in article is completed as response#1. The literatures given above really have discussed the nanoindentation mapping results in detail. In our article, the areas of the nanoindentation mapping and SEM images are different. The range of mapping area is 150 µm×150 µm, under this size scale the grains and indentation could not be clearly seen by SEM. So we could not combine the mapping and SEM images together at this moment, we may solute this problem in our following research. The size of grains is smaller than 5μm and even the nanoscale, it is hard to distinguish three phase just by visual inspection. So the statistic data is shown in the Figure 6. The hardness and modulus of different phases changing with the annealing time are discussed and modified in the section 3.2.

Point #3 I do not understand figure 3. The 3c and 3d is perhaps the magnified view from 3a but the chosen area is not highlighted. 3b have different %ages on the figure. What different colour coding means, it is not highlighted in the caption. This should be better presented!

Response#3 Original Figure.3 changes to Figure.1 in section 3.1. In order to provide abundant evidence, Figure 1c and 1d are another two different areas, different from Figure 1a area. Figure 1b is the quantitative analysis of EDS mapping. Different colour coding means various weight of Y element, added in the article.

Point #4 Mechanical properties of single phases (figure 6 and 7, perhaps combine both images?) is very interesting, however discussion should be made about the standard deviation. It is worth highlighting here again that how many indentations were performed and why the standard deviation overlaps for 9, 12 and 24 hours. Detail discussion should be made about the trend.

Response#4 Thanks for your suggestion. Figure 6 and Figure 7 change to Figure 7 and Figure8, and they have been put together in the article. In our research, ten nanoindentation tests of individual phase are performed. Besides, there are standard deviation overlaps for 9, 12 and 24 hours particularly for T′ and C phase, this is because the uncertainty of indentation position during test, and some value may come from the grain boundary of Tand C phase. The trends in Figure 8 are discussed in detail in revised section 3.3.

Conclusions

Point Based on the above-mentioned changes, this section needs to be re-written with more clarity that how the nanoindentation mapping help distinguish between different phases and provide reliable measurement of both hardness and modulus of different phases.

Response The conclusions have been re-written. Please see the modifications in the article.

General Remarks

Point Technical English needs improvement. There are some grammatical mistakes which authors should carefully check.

Response Thank you so much. Grammatical mistakes have been fixed carefully.

Reviewer 2 Report

The authors have demonstrated that nanoindentation can be used to identify mechanical property changes in t’ YSZ caused by thermal degradation. They have provided XRD and EDS to support the mechanical performance change is due to phase transformations because of the phase’s instability at high temperatures. However, the text could be improved by clarifying some of the methods and providing some additional supporting evidence for their conclusions. Please read below for specific questions the authors need to address.

1)    The authors cite that a major reason for this technique is to provide mechanical performance of individual phases. How is hardness important to the application sited (TBCs)? Additionally, was there any analysis to see how individual phases contributed to the bulk performance, as indicated in the introduction?

2)    Nanoindentation has been used to identify the regions where the t’-> m phase transformation occurred in YSZ previously. The authors should include these references and clarify statements regarding the novelty of this technique.

3)    Using this nanoindentation mapping method, was there any checks to ensure that observed property changes were not due to hitting grain boundaries or from changes in microstructure?

4)    Methods: could the authors expand on what is different between the mapping mode and the CSM technique?

5)    For Figure 4: Please state clearly how these phases were identified in the methods. Was this same area tested with nanoindentation for direct comparison?

6)    Figure 5: How was the nanoindentation map quantified to make this figure? Please provide description for how each phase was identified in the mapping (what were the limits for each mechanical property used to delineate phases?) Also, please see comment #9.

7)    Figure 5: Please edit caption to include that data in b was from XRD.

8)    Section 3.3: You conclude there was insufficient cubic phase at 2 hrs and only include hardness data from >9 hrs. However, the XRD data in Figure 5 suggests there is not that much change in the cubic phase from 2 to 9 hours. Please explain this inconsistency.

9)    Figure 7: Given the similarity in hardness between C and t’ – please describe the feasibility of this technique to reliably distinguish these phases.

10) Figure 7: Was EDS used to parse out these phases during this part of the mechanical testing? Please provide more description in the methods to indicate how phase was identified.

11) In the discussion, you suggest that the increase in t’ phase is due to an increase in Y content. Was this measured with EDS or another technique? Supporting evidence should be provided or citations indicating a similar behavior.  

Author Response

Dear reviewer:

On behalf of my co-authors, we thank you very much for giving us an opportunity to revise our manuscript. We appreciate for your comments and suggestions on the manuscript entitled “Evaluation of Phase Transformation and Mechanical Properties of Metastable yttria-stabilized zirconia by Nanoindentation (ID: materials-498383)”. We have read the comments carefully and tried our best to revise the manuscript accordingly. Revised portion are highlighted with red words in the paper. The main corrections in the paper and the responses to referee’s comments are as following.

Response to Reviewer

Point#1 The authors cite that a major reason for this technique is to provide mechanical performance of individual phases. How is hardness important to the application sited (TBCs)? Additionally, was there any analysis to see how individual phases contributed to the bulk performance, as indicated in the introduction?

Response#1 Thank you for your comment. The nanoindentation is a facilitate method to test hardness and Young’s modulus of phases and grains at nanoscale. The hardness and Young’s modulus are the basic properties in TBCs and have effect on its durability. Indeed, the bulk performance is related with the character of individual phases. In present report, we adopted the nanoindentation method to investigate the phase transition along with high temperature aging. The nanoindentation method could distinguish the hardness and Young’s modulus of individual grain at nano scale, but is hard to test the strength and toughness and other mechanical property. In the future research, we would try to build a correlation between individual phases and bulk performance. We revise the manuscript according to your comment.

Point #2 Nanoindentation has been used to identify the regions where the t’-> m phase transformation occurred in YSZ previously. The authors should include these references and clarify statements regarding the novelty of this technique.

Response#2 Thanks for your advice. The novelty of this technique citing relevant literature in YSZ is added in the introduction.

Point #3 Using this nanoindentation mapping method, was there any checks to ensure that observed property changes were not due to hitting grain boundaries or from changes in microstructure?

Response#3 Nanoindentation mapping method really could display the minor change of mechanical properties on the sample surface. I’m so sorry to say that we could not make sure that all the property changes are not due to hitting grain boundaries or from changes in microstructure by now.    

Point #4 Methods: could the authors expand on what is different between the mapping mode and the CSM technique?

Response#4 Thanks for your suggestion. The manuscript is modified in section 2.3.

Point #5 For Figure 4: Please state clearly how these phases were identified in the methods. Was this same area tested with nanoindentation for direct comparison?

Response#5 Sorry for being unclear about the phases identification (Figure 4 change to Figure 2 in the article). According to the EDS results, the lean Y grains are T phase, the rich Y grains are C phase, the Intermediate Y content is T′. The area of SEM images is inside the nanoindentation mapping. However, The area of mapping is 150 µm × 150 µm, nearly 10 times of SEM images, and we cannot point out the position of SEM observation area in the nanoindentation mapping because the operation was conducted in two instruments individually.

Point #6 Figure 5: How was the nanoindentation map quantified to make this figure? Please provide description for how each phase was identified in the mapping (what were the limits for each mechanical property used to delineate phases?) Also, please see comment #9.

Response#6 (Figure 5 change to Figure 6 in the article) The data are processed by Matlab software. The criteria are identified by the hardness value or corresponding colour coding at different annealing time. Meanwhile, the change in mechanical properties for individual phase is also to be considered.

Point #7 Figure 5: Please edit caption to include that data in b was from XRD.

Response#7 Thanks for your suggestion. Caption has been edited in Figure 6(original Figure 5).

Point #8 Section 3.3: You conclude there was insufficient cubic phase at 2 hrs and only include hardness data from >9 hrs. However, the XRD data in Figure 5 suggests there is not that much change in the cubic phase from 2 to 9 hours. Please explain this inconsistency.

Response#8 Although the increase of C phase volume content is not significant from 2 to 9 hours aging, the grain growth of C phase grain is very distinct and easy to be detected by the nanoindentation after 9 hours aging, compared with the small grains C phase grains formed during first 2h aging.

Point #9 Figure 7: Given the similarity in hardness between C and t’ – please describe the feasibility of this technique to reliably distinguish these phases.

Response#9 Indeed, the hardness value of C is close to T′ phase, and it is difficult to distinguish them. We have to measure by means of EDS. It is easy to differentiate T phase grain with T′ and C phases by nanoindentation. There is still space to improve the validity in distinguish transition using nanoindentation system in future.  

Point #10 Figure 7: Was EDS used to parse out these phases during this part of the mechanical testing? Please provide more description in the methods to indicate how phase was identified.

Response#10 As an auxiliary means, EDS was used to parse out these phases due to Y element content. But it is impossible to detect EDS during mechanical testing. Detailed description was added in the revised paper, section 3.1.

Point #11 In the discussion, you suggest that the increase in t’ phase is due to an increase in Y content. Was this measured with EDS or another technique? Supporting evidence should be provided or citations indicating a similar behavior.  

Response#11 May some misunderstanding caused by our manuscript, no “increase in t′ phase” described in our report, only “decrease”. The t′ phase is a metastable phase formed by APS or EB-PVD process, and will decompose at high temperature accompanied with Y diffusion. The EDS result can distinguish Y content in different phase grains very clearly. The paper has revised.

Reviewer 3 Report

Overall, this is good work.  I do think there are some improvements that will significantly strengthen your conclusions from the study... please see my comments attached to your manuscript.

Also, I HIGHLY recommend proof-reading by a native English speaker.  This would improve the readability of the manuscript.

Author Response

Dear reviewer:

On behalf of my co-authors, we thank you very much for giving us an opportunity to revise our manuscript. We appreciate for your comments and suggestions on the manuscript entitled “Evaluation of Phase Transformation and Mechanical Properties of Metastable yttria-stabilized zirconia by Nanoindentation (ID: materials-498383)”. We have read the comments carefully and tried our best to revise the manuscript accordingly. Revised portion are highlighted with red words in the paper. The main corrections in the paper and the responses to referee’s comments are as following.

Response to the reviewer

Point#1 How do you know these are T' specimens?

Response#1 Thank you for your comment. T' specimens has been estimated by XRD. It is added in section2.1 to mention XRD results. This is a well known method.

Point #2 did you measure an as-prepared sample? what is the hardness distribution of the as-prepared sample?

Response#2 It is a good idea. I am sorry to say that there is no other results about as-prepared sample. In the future work, I would try to make comparison scrupulously.

Point #3 this is a common issue with auto-correct... please correct - units should be GPa not Gpa.

Response#3 All the mistakes have been fixed.

Point #4 be careful with your reporting.... you cannot definitively claim to know the oxidation state from only an EDS measurement (which only reveals the elemental distribution).

Response#4 Thanks for your reminding. I already corrected the Y oxidation state to Y element.

Point #5 it would be EXTREMELY helpful to see the nanoindentation mapping results plotted next to the SEM+EDS results... the readers would be able to SEE the correlation for themselves!

NOTE: while useful, this does not need to be done for each annealing case... I suggest plotting an as-prepared sample and the 24 h annealing time sample.

Response#5 Thanks for your advice. It is really a good idea. In our article, the area SEM images is too smaller than nanoindentation mapping because we want to observe the grain morphology clearly, and the same to EDS. On the other hand, the range of mapping is 150 µm×150 µm, as we need some statistical results. So the nanoindentation mapping could not be plotted next to the SEM+EDS results, as the size difference of them is too larger. We will try to solute this in next step research.

Point #6 what is your residuals from the peak-fitting?  it may be helpful to plot the XRD patterns and fitted results (with some inidication of the residuals / fit quality) as a separate figure rather than an inset.

Response#6 Figure 5 The evolution of XRD patterns and five diffraction peaks of C(004), T(004), T(400), T′(004) and T′(400) by Lorentz peak fitting has been added. The residuals from the peak-fitting has been added in section 3.3. Please see the modification in the article.

Round  2

Reviewer 1 Report

It is in a much better shape, however i will still recommend few changes. 

1/ Abstract should also reflect the results. It is more like an introduction and provides no values at all. Authors claim that this is the first time the mechanical properties of different phases are measured. This should be highlighted in the abstract. Please also add some numbers along with the description that how the H and E changes for different phases. 

2/ English needs to be checked once again. There are plenty of grammatical mistakes and in some cases the sentences do not make any sense. Few examples are as follows:  

Page 2: "Recently, measurements of mechanical properties for a wide range of materials become available by a versatile nanoindentation technique" .... mechanical properties are not available because of nanoindentation technique. Please rephrase!

Page 4: "phase transformation occurs, followed by the changes in mechanical properties of 8YSZ".... phase transformation results in change of mechanical properties 

Page 9: "was measured by a nonoindentation system" ... just nanoindentation is fine!

Page 9: "It is demonstrated that C phase has higher hardness and Young’s modulus than T′ phase and the T phase shows the least value among the three, which is consistent with the sequence of Y content in these phases." ... needs rephrasing

Page 9: "aging time could be evaluated by nonoindentation mapping" ... you are concluding your work. If you say could be evaluated that means you are introducing a doubt in your own work. Please rephrase 

Author Response

Dear reviewer:

On behalf of my co-authors, we thank you very much for giving us an opportunity to revise our manuscript. We appreciate for your comments and suggestions on the manuscript entitled “Evaluation of Phase Transformation and Mechanical Properties of Metastable yttria-stabilized zirconia by Nanoindentation (ID: materials-498383)”. We have read the comments carefully and tried our best to revise the manuscript accordingly. Revised portion are highlighted with blue words in the paper. The main corrections in the paper and the responses to referee’s comments are as following.

Response to the reviewer

Point 1 Abstract should also reflect the results. It is more like an introduction and provides no values at all. Authors claim that this is the first time the mechanical properties of different phases are measured. This should be highlighted in the abstract. Please also add some numbers along with the description that how the H and E changes for different phases. 

Response 1 The abstract have been re-written. Please see the modifications in the article.

Point 2English needs to be checked once again. There are plenty of grammatical mistakes and in some cases the sentences do not make any sense. Few examples are as follows:  

Page 2: "Recently, measurements of mechanical properties for a wide range of materials become available by a versatile nanoindentation technique" .... mechanical properties are not available because of nanoindentation technique. Please rephrase!

Page 4: "phase transformation occurs, followed by the changes in mechanical properties of 8YSZ".... phase transformation results in change of mechanical properties 

Page 9: "was measured by a nonoindentation system" ... just nanoindentation is fine!

Page 9: "It is demonstrated that C phase has higher hardness and Young’s modulus than T′ phase and the T phase shows the least value among the three, which is consistent with the sequence of Y content in these phases." ... needs rephrasing

Page 9: "aging time could be evaluated by nonoindentation mapping" ... you are concluding your work. If you say could be evaluated that means you are introducing a doubt in your own work. Please rephrase 

Response 2 The language has been checked once more. The grammatical mistakes have been revised.

Reviewer 3 Report

please see the comments (i.e. "sticky notes") embedded in the attached pdf document of your manuscript.

I still believe that it is essential that you present information / results of testing on the as-prepared samples.  For example, why did you present a graphic of the as-prepared XRD in your response BUT not the paper?  This graphic, along with analysis of the phase content, would be greatly appreciated by the readers.  In addition, the lack of nanoindentation results for the as-prepared samples is a clear lack of a control in your methodology. 

In addition, I do not accept the argument on the size differences are too great to accurately plot the SEM-EDS results and nanoindentation maps side-by-side.  Figure 7 clearly shows multiple phases are present with fifteen (15) nanoindentation points!  Therefore, it is really only a matter of properly identifying the area of interest AND scaling the nanoindentation map appropriately.

Author Response

Dear reviewer:

On behalf of my co-authors, we thank you very much for giving us an opportunity to revise our manuscript. We appreciate for your comments and suggestions on the manuscript entitled “Evaluation of Phase Transformation and Mechanical Properties of Metastable yttria-stabilized zirconia by Nanoindentation (ID: materials-498383)”. We have read the comments carefully and tried our best to revise the manuscript accordingly. Revised portion are highlighted with blue words in the paper. The main corrections in the paper and the responses to referee’s comments are as following.

Response to the reviewer 

Point 1 define m phase, as you did with the tetragonal (T) and cubic (C) phases

Response 1 M phase is defined in the article.

Point 2where is the data to support this? what was the estimated phase content in the as-prepared samples?

Response 2 The X-ray diffraction result of as-prepared sample is shown in the figure 5. Please see the modification in the article. As-prepared sample contains T’ phase only.

Point 3 how do you know which grain is which?  you are making an assumption, based on prior experience and previous literature... in other words, "The fine grains, with lean Y content, are most likely T phase and the coarse grains, with rich Y content, are most likely to be C phase.  This would be consistent with..."

Definitive statements are VERY hard to come by in science... a lot of evidence needs to be gathered and assessed before a DEFINITIVE statement can be made.

One key tool that could be used for definitive phase identification is HR-TEM, coupled with SAED.  This would produce the necessary evidence to definitively say this grain is T phase and this grain is C phase.

Response 3 Thanks for your suggestions. As the Y content in T′, T and C phases are very different and the EDS is an effective method to distinct these phases on SEM, the relevant literatures are shown below. This method is widely used by researchers to study the structures of YSZ ceramics or coatings. XRD is another technique to determine the phase content. HR-TEM and SAED are good methods to provide evidence to identify the phases. But it is a pity that we cannot prepare the high quality specimen for HR-TEM analysis at present. In the future research, we would try to adopt the HR-TEM and SAED in our phase transition research.

1.        Yao Hu, Canying Cai, Yanguo Wang, Hongchun Yu, Yichun Zhou, Guangwen Zhou, YSZ/NiCrAlY interface oxidation of APS thermal barrier coatings, Corrosion Science, Volume 142, 2018, Pages 22-30, ISSN 0010-938X, https://doi.org/10.1016/j.corsci.2018.06.035.

2.        Jinshuang Wang, Junbin Sun, Jieyan Yuan, Qiangshan Jing, Shujuan Dong, Bing Liu, Hao Zhang, Longhui Deng, Jianing Jiang, Xin Zhou, Xueqiang Cao, Phase stability, thermo-physical properties and thermal cycling behavior of plasma-sprayed CTZ, CTZ/YSZ thermal barrier coatings, Ceramics International, Volume 44, Issue 8, 2018, Pages 9353-9363, ISSN 0272-8842, https://doi.org/10.1016/j.ceramint.2018.02.149.

3.      Xiaorui Ren, Wei Pan, Mechanical properties of high-temperature-degraded yttria-stabilized zirconia, Acta Materialia, Volume 69, 2014, Pages 397-406, ISSN 1359-6454, https://doi.org/10.1016/j.actamat.2014.01.017.

Point 4 I still believe that it is essential that you present information / results of testing on the as-prepared samples.  For example, why did you present a graphic of the as-prepared XRD in your response BUT not the paper?  This graphic, along with analysis of the phase content, would be greatly appreciated by the readers.  In addition, the lack of nanoindentation results for the as-prepared samples is a clear lack of a control in your methodology. 

Response 4 The X-ray diffraction result of as-prepared sample is added in the figure 5. I feel very sorry that I cannot meet your requirement, because the quality of as-prepared samples is poor when we do the nanoindentation testing. It is known that the condition of sample for nanoindentation mapping is strict. The porosity of the as-prepared sample was quit high when we take all the sample to the nanoindentation testing. We found that the nanoindentation mapping results could not be used and now it is difficult to prepare the sample again with no aging.

Point 5 From figure 7a, we can see that the SEM image contains 15 indents across three different phases... why can you not show the nanoindentation map (for this area) with this SEM image?  it is only a matter of identifying the appropriate area on the  nanoindentation map AND scaling appropriately.

Response 5 Thanks for your suggestion. We tried our best to display the SEM image corresponding to nanoindentation mapping. In figure 7a, the indenters are obtained by CSM mode of nanoindentation showing clear imagine of trace of indentation. The depth of them are 100nm. However, the depth of indenters obtained by nanoindentation mapping is shallower, due to its high speed scan mode foe mapping. The nanoindentation mapping and SEM image are shown below. The trace of indentation are not clear. So we did not show this result in the article.

Point 6 the variation / enhancement does not appear to be statistically significant, given the size of your error bars.

Response 6 Indeed, the error bars are a little bit large in present work. This is because there is uncertainty of indentation position during test, and some value may come from the grain boundary of T′ and C phase, or some pores on the surface of specimen. In the future, we will improve the surface quality of sample and indentation technology to ensure the indenter right on the grains.

Point 7 is this reasoning supported by EDS results from the sample annealed for 9 hours?

Response 7 It was not by the EDS! It was just estimated based on the discussion in the article. In future research, EDS and other analysis will be added in the following work. We revised the part accordingly.